# FREE LUNCH FOR FEW-SHOT LEARNING: DISTRIBUTION CALIBRATION

**Shuo Yang**[1]**, Lu Liu**[2]**, Min Xu**[1][*]
[1]School of Electrical and Data Engineering, University of Technology Sydney,
[2]Australian Artificial Intelligence Institute, University of Technology Sydney
{shuo.yang, lu.liu-10}@student.uts.edu.au, min.xu@uts.edu.au,

## ABSTRACT

Learning from a limited number of samples is challenging since the learned model can easily become overfitted based on the biased distribution formed by only a few training examples. In this paper, we calibrate the distribution of these few-sample classes by transferring statistics from the classes with sufficient examples. Then an adequate number of examples can be sampled from the calibrated distribution to expand the inputs to the classifier. We assume every dimension in the feature representation follows a Gaussian distribution so that the mean and the variance of the distribution can borrow from that of similar classes whose statistics are better estimated with an adequate number of samples. Our method can be built on top of off-the-shelf pretrained feature extractors and classification models without extra parameters. We show that a simple logistic regression classifier trained using the features sampled from our calibrated distribution can outperform the state-of-the-art accuracy on three datasets (5% improvement on miniImageNet compared to the next best). The visualization of these generated features demonstrates that our calibrated distribution is an accurate estimation. The code is available at: https://github.com/ShuoYang-1998/Few_Shot_Distribution_Calibration

## 1 INTRODUCTION

Learning from a limited number of training samples has drawn increasing attention due to the high cost of collecting and annotating a large amount of data. Researchers have developed algorithms to improve the performance of models that have been trained with very few data. Finn et al. (2017); Snell et al. (2017) train models in a meta-learning fashion so that the model can adapt quickly on tasks with only a few training samples available. Hariharan & Girshick (2017); Wang et al. (2018) try to synthesize data or features by learning a generative model to alleviate the data insufficiency problem. Ren et al. (2018) propose to leverage unlabeled data and predict pseudo labels to improve the performance of few-shot learning.

Table 1: The class mean similarity ("mean sim") and class variance similarity ("var sim") between Arctic fox and different classes.

| | Arctic fox | |
|---|---|---|
| | mean sim | var sim |
| white wolf | 97% | 97% |
| malamute | 85% | 78% |
| lion | 81% | 70% |
| meerkat | 78% | 70% |
| jellyfish | 46% | 26% |
| orange | 40% | 19% |
| beer bottle | 34% | 11% |

While most previous works focus on developing stronger models, scant attention has been paid to the property of the data itself. It is natural that when the number of data grows, the ground truth distribution can be more accurately uncovered. Models trained with a wide coverage of data can generalize well during evaluation. On the other hand, when training a model with only a few training data, the model tends to overfit on these few samples by minimizing the training loss over these samples. These phenomena are illustrated in Figure 1. This biased distribution based on a few examples can damage the generalization ability of the model since it is far from mirroring the ground truth distribution from which test cases are sampled during evaluation.

---

[*]Corresponding author.

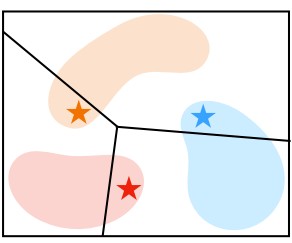
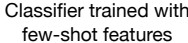
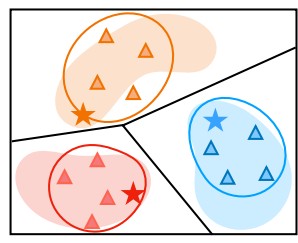
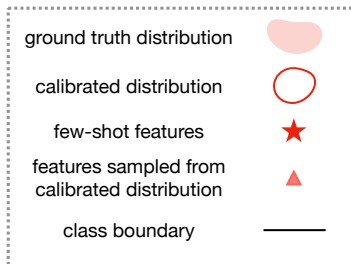

Figure 1: Training a classifier from few-shot features makes the classifier overfit to the few examples (Left). Classifier trained with features sampled from calibrated distribution has better generalization ability (Right).

Here, we consider calibrating this biased distribution into a more accurate approximation of the ground truth distribution. In this way, a model trained with inputs sampled from the calibrated distribution can generalize over a broader range of data from a more accurate distribution rather than only fitting itself to those few samples. Instead of calibrating the distribution of the original data space, we try to calibrate the distribution in the feature space, which has much lower dimensions and is easier to calibrate (Xian et al. (2018)). We assume every dimension in the feature vectors follows a Gaussian distribution and observe that similar classes usually have similar mean and variance of the feature representations, as shown in Table 1. Thus, the mean and variance of the Gaussian distribution can be transferred across similar classes (Salakhutdinov et al. (2012)). Meanwhile, the statistics can be estimated more accurately when there are adequate samples for this class. Based on these observations, we reuse the statistics from many-shot classes and transfer them to better estimate the distribution of the few-shot classes according to their class similarity. More samples can be generated according to the estimated distribution which provides sufficient supervision for training the classification model.

In the experiments, we show that a simple logistic regression classifier trained with our strategy can achieve state-of-the-art accuracy on three datasets. Our distribution calibration strategy can be paired with any classifier and feature extractor with no extra learnable parameters. Training with samples selected from the calibrated distribution can achieve 12% accuracy gain compared to the baseline which is only trained with the few samples given in a 5way1shot task. We also visualize the calibrated distribution and show that it is an accurate approximation of the ground truth that can better cover the test cases.

## 2 RELATED WORKS

Few-shot classification is a challenging machine learning problem and researchers have explored the idea of learning to learn or meta-learning to improve the quick adaptation ability to alleviate the few-shot challenge. One of the most general algorithms for meta-learning is the optimization-based algorithm. Finn et al. (2017) and Li et al. (2017) proposed to learn how to optimize the gradient descent procedure so that the learner can have a good initialization, update direction, and learning rate. For the classification problem, researchers proposed simple but effective algorithms based on metric learning. MatchingNet (Vinyals et al., 2016) and ProtoNet (Snell et al., 2017) learned to classify samples by comparing the distance to the representatives of each class. Our distribution calibration and feature sampling procedure does not include any learnable parameters and the classifier is trained in a traditional supervised learning way.

Another line of algorithms is to compensate for the insufficient number of available samples by generation. Most methods use the idea of Generative Adversarial Networks (GANs) (Goodfellow et al., 2014) or autoencoder (Rumelhart et al., 1986) to generate samples (Zhang et al. (2018); Chen et al. (2019b); Schwartz et al. (2018); Gao et al. (2018)) or features (Xian et al. (2018); Zhang et al. (2019)) to augment the training set. Specifically, Zhang et al. (2018) and Xian et al. (2018) proposed to synthesize data by introducing an adversarial generator conditioned on tasks. Zhang et al. (2019) tried to learn a variational autoencoder to approximate the distribution and predict labels based on the estimated statistics. The autoencoder can also augment samples by projecting between the visual

space and the semantic space (Chen et al., 2019b) or encoding the intra-class deformations (Schwartz et al., 2018). Liu et al. (2019b) and Liu et al. (2019a) propose to generate features through the class hierarchy. While these methods can generate extra samples or features for training, they require the design of a complex model and loss function to learn how to generate. However, our distribution calibration strategy is simple and does not need extra learnable parameters.

Data augmentation is a traditional and effective way of increasing the number of training samples. Qin et al. (2020) and Antoniou & Storkey (2019) proposed the used of the traditional data augmentation technique to construct pretext tasks for unsupervised few-shot learning. Wang et al. (2018) and Hariharan & Girshick (2017) leveraged the general idea of data augmentation, they designed a hallucination model to generate the augmented version of the image with different choices for the model's input, i.e., an image and a noise  (Wang et al., 2018) or the concatenation of multiple features (Hariharan & Girshick, 2017).  Park et al. (2020); Wang et al. (2019); Liu et al. (2020b) tried to augment feature representations by leveraging intra-class variance. These methods learn to augment from the original samples or their feature representation while we try to estimate the class-level distribution and thus can eliminate the inductive bias from a single sample and provide more diverse generations from the calibrated distribution.

# 3 MAIN APPROACH

In this section, we introduce the few-shot classification problem definition in Section 3.1 and details of our proposed approach in Section 3.2.

## 3.1 PROBLEM DEFINITION

We follow a typical few-shot classification setting. Given a dataset with data-label pairs $\mathcal{D} = \{(\boldsymbol{x}_i, y_i)\}$ where $\boldsymbol{x}_i \in \mathbb{R}^d$ is the feature vector of a sample and $y_i \in C$, where $C$ denotes the set of classes. This set of classes is divided into *base classes* $C_b$ and *novel classes* $C_n$, where $C_b \cap C_n = \emptyset$ and $C_b \cup C_n = C$. The goal is to train a model on the data from the base classes so that the model can generalize well on tasks sampled from the novel classes. In order to evaluate the fast adaptation ability or the generalization ability of the model, there are only a few available labeled samples for each task $\mathcal{T}$. The most common way to build a task is called an N-way-K-shot task (Vinyals et al. (2016)), where N classes are sampled from the novel set and only K (e.g., 1 or 5) labeled samples are provided for each class. The few available labeled data are called *support set* $\mathcal{S} = \{(\boldsymbol{x}_i, y_i)\}_{i=1}^{N \times K}$ and the model is evaluated on another query set $\mathcal{Q} = \{(\boldsymbol{x}_i, y_i)\}_{i=N \times K+1}^{N \times K + N \times q}$, where every class in the task has $q$ test cases. Thus, the performance of a model is evaluated as the averaged accuracy on (the query set of) multiple tasks sampled from the novel classes.

## 3.2 DISTRIBUTION CALIBRATION

As introduced in Section 3.1, the base classes have a sufficient amount of data while the evaluation tasks sampled from the novel classes only have a limited number of labeled samples. The statistics of the distribution for the base class can be estimated more accurately compared to the estimation based on few-shot samples, which is an ill-posed problem. As shown in Table 1, we observe that if we assume the feature distribution is Gaussian, the mean and variance with respect to each class are correlated to the semantic similarity of each class. With this in mind, the statistics can be transferred from the base classes to the novel classes if we learn how similar the two classes are. In the following sections, we discuss how we calibrate the distribution estimation of the classes with only a few samples (Section 3.2.2) with the help of the statistics of the base classes (Section 3.2.1). We will also elaborate on how do we leverage the calibrated distribution to improve the performance of few-shot learning (Section 3.2.3).

Note that our distribution calibration strategy is over the feature-level and is agnostic to any feature extractor. Thus, it can be built on top of any pretrained feature extractors without further costly fine-tuning. In our experiments, we use the pretrained WideResNet Zagoruyko & Komodakis (2016) following previous work (Mangla et al. (2020)). The WideResNet is trained to classify the base classes, along with a self-supervised pretext task to learn the general-purpose representations suitable for image understanding tasks. Please refer to their paper for more details on training the feature extractor.

---

**Algorithm 1** Training procedure for an N-way-K-shot task

---

**Require:** Support set features $\mathcal{S} = (\boldsymbol{x}_i, y)_{i=1}^{N \times K}$
**Require:** Base classes' statistics $\{\boldsymbol{\mu}_i\}_{i=1}^{|C_b|}, \{\boldsymbol{\Sigma}_i\}_{i=1}^{|C_b|}$
 1: Transform $(\boldsymbol{x}_i)_{i=1}^{N \times K}$ with Tukey's Ladder of Powers as Equation 3
 2: **for** $(\boldsymbol{x}_i, y_i) \in \mathcal{S}$ **do**
 3:     Calibrate the mean $\boldsymbol{\mu}'$ and the covariance $\boldsymbol{\Sigma}'$ for class $y_i$ using $\boldsymbol{x}_i$ with Equation 6
 4:     Sample features for class $y_i$ from the calibrated distribution as Equation 7
 5: **end for**
 6: Train a classifier using both support set features and all sampled features as Equation 8

---

### 3.2.1 STATISTICS OF THE BASE CLASSES

We assume the feature distribution of base classes is Gaussian. The mean of the feature vector from a base class $i$ is calculated as the mean of every single dimension in the vector:

$$\boldsymbol{\mu}_i = \frac{\sum_{j=1}^{n_i} \boldsymbol{x}_j}{n_i}, \tag{1}$$

where $\boldsymbol{x}_j$ is a feature vector of the $j$-th sample from the base class $i$ and $n_i$ is the total number of samples in class $i$. As the feature vector $\boldsymbol{x}_j$ is multi-dimensional, we use covariance for a better representation of the variance between any pair of elements in the feature vector. The covariance matrix $\boldsymbol{\Sigma}_i$ for the features from class $i$ is calculated as:

$$\boldsymbol{\Sigma}_i = \frac{1}{n_i - 1} \sum_{j=1}^{n_i} (\boldsymbol{x}_j - \boldsymbol{\mu}_i)(\boldsymbol{x}_j - \boldsymbol{\mu}_i)^T. \tag{2}$$

### 3.2.2 CALIBRATING STATISTICS OF THE NOVEL CLASSES

Here, we consider an N-way-K-shot task sampled from the novel classes.

**Tukey's Ladder of Powers Transformation**

To make the feature distribution more Gaussian-like, we first transform the features of the support set and query set in the target task using Tukey's Ladder of Powers transformation (Tukey (1977)). Tukey's Ladder of Powers transformation is a family of power transformations which can reduce the skewness of distributions and make distributions more Gaussian-like. Tukey's Ladder of Powers transformation is formulated as:

$$\tilde{\mathbf{x}} = \begin{cases} \boldsymbol{x}^\lambda & \text{if } \lambda \neq 0 \\ \log(\boldsymbol{x}) & \text{if } \lambda = 0 \end{cases} \tag{3}$$

where $\lambda$ is a hyper-parameter to adjust how to correct the distribution. The original feature can be recovered by setting $\lambda$ as 1. Decreasing $\lambda$ makes the distribution less positively skewed and vice versa.

**Calibration through statistics transfer**

Using the statistics from the base classes introduced in Section 3.2.1, we transfer the statistics from the base classes which are estimated more accurately on sufficient data to the novel classes. The transfer is based on the Euclidean distance between the feature space of the novel classes and the mean of the features from the base classes $\boldsymbol{\mu}_i$ as computed in Equation 1. Specifically, we select the top $k$ base classes with the closest distance to the feature of a sample $\tilde{\boldsymbol{x}}$ from the support set:

$$\mathbb{S}_d = \{-\|\boldsymbol{\mu}_i - \tilde{\boldsymbol{x}}\|^2 \mid i \in C_b\}, \tag{4}$$

$$\mathbb{S}_N = \{i \mid -\|\boldsymbol{\mu}_i - \tilde{\boldsymbol{x}}\|^2 \in topk(\mathbb{S}_d)\}, \tag{5}$$

where $topk(\cdot)$ is an operator to select the top elements from the input distance set $\mathbb{S}_d$. $\mathbb{S}_N$ stores the $k$ nearest base classes with respect to a feature vector $\tilde{\boldsymbol{x}}$. Then, the mean and covariance of the distribution is calibrated by the statistics from the nearest base classes:

$$\boldsymbol{\mu}' = \frac{\sum_{i \in \mathbb{S}_N} \boldsymbol{\mu}_i + \tilde{\boldsymbol{x}}}{k+1}, \boldsymbol{\Sigma}' = \frac{\sum_{i \in \mathbb{S}_N} \boldsymbol{\Sigma}_i}{k} + \alpha, \tag{6}$$

Table 2: 5way1shot and 5way5shot classification accuracy (%) on *mini*ImageNet and CUB with 95% confidence intervals. The numbers in **bold** have intersecting confidence intervals with the most accurate method.

| Methods | *mini*ImageNet | | CUB | |
|---|---|---|---|---|
| | 5way1shot | 5way5shot | 5way1shot | 5way5shot |
| ***Optimization-based*** | | | | |
| MAML (Finn et al. (2017)) | $48.70 \pm 1.84$ | $63.10 \pm 0.92$ | $50.45 \pm 0.97$ | $59.60 \pm 0.84$ |
| Meta-SGD (Li et al. (2017)) | $50.47 \pm 1.87$ | $64.03 \pm 0.94$ | $53.34 \pm 0.97$ | $67.59 \pm 0.82$ |
| LEO (Rusu et al. (2019)) | $61.76 \pm 0.08$ | $77.59 \pm 0.12$ | - | - |
| E3BM (Liu et al. (2020c)) | $63.80 \pm 0.40$ | $80.29 \pm 0.25$ | - | - |
| ***Metric-based*** | | | | |
| Matching Net (Vinyals et al. (2016)) | $43.56 \pm 0.84$ | $55.31 \pm 0.73$ | $56.53 \pm 0.99$ | $63.54 \pm 0.85$ |
| Prototypical Net (Snell et al. (2017)) | $54.16 \pm 0.82$ | $73.68 \pm 0.65$ | $72.99 \pm 0.88$ | $86.64 \pm 0.51$ |
| Baseline++ (Chen et al. (2019a)) | $51.87 \pm 0.77$ | $75.68 \pm 0.63$ | $67.02 \pm 0.90$ | $83.58 \pm 0.54$ |
| Variational Few-shot(Zhang et al. (2019)) | $61.23 \pm 0.26$ | $77.69 \pm 0.17$ | - | - |
| Negative-Cosine(Liu et al. (2020a)) | $62.33 \pm 0.82$ | $80.94 \pm 0.59$ | $72.66 \pm 0.85$ | $89.40 \pm 0.43$ |
| ***Generation-based*** | | | | |
| MetaGAN (Zhang et al. (2018)) | $52.71 \pm 0.64$ | $68.63 \pm 0.67$ | - | - |
| Delta-Encoder (Schwartz et al. (2018)) | $59.9$ | $69.7$ | $69.8$ | $82.6$ |
| TriNet (Chen et al. (2019b)) | $58.12 \pm 1.37$ | $76.92 \pm 0.69$ | $69.61 \pm 0.46$ | $84.10 \pm 0.35$ |
| Meta Variance Transfer (Park et al. (2020)) | - | $67.67 \pm 0.70$ | - | $80.33 \pm 0.61$ |
| Maximum Likelihood with DC (Ours) | $66.91 \pm 0.17$ | $80.74 \pm 0.48$ | $77.22 \pm 0.14$ | $89.58 \pm 0.27$ |
| SVM with DC (Ours) | $\mathbf{67.31 \pm 0.83}$ | $\mathbf{82.30 \pm 0.34}$ | $\mathbf{79.49 \pm 0.33}$ | $\mathbf{90.26 \pm 0.98}$ |
| Logistic Regression with DC (Ours) | $\mathbf{68.57 \pm 0.55}$ | $\mathbf{82.88 \pm 0.42}$ | $\mathbf{79.56 \pm 0.87}$ | $\mathbf{90.67 \pm 0.35}$ |

where $\alpha$ is a hyper-parameter that determines the degree of dispersion of features sampled from the calibrated distribution.

For few-shot learning with more than one shot, the aforementioned procedure of the distribution calibration should be undertaken multiple times with each time using one feature vector from the support set. This avoids the bias provided by one specific sample and potentially achieves more diverse and accurate distribution estimation. Thus, for simplicity, we denote the calibrated distribution as a set of statistics. For a class $y \in C_n$, we denote the set of statistics as $\mathbb{S}_y = \{(\boldsymbol{\mu}_1', \boldsymbol{\Sigma}_1'), ..., (\boldsymbol{\mu}_K', \boldsymbol{\Sigma}_K')\}$, where $\boldsymbol{\mu}_i', \boldsymbol{\Sigma}_i'$ are the calibrated mean and covariance, respectively, computed based on the $i$-th feature in the support set of class $y$. Here, the size of the set is the value of $K$ for an N-way-K-shot task.

### 3.2.3 HOW TO LEVERAGE THE CALIBRATED DISTRIBUTION?

With a set of calibrated statistics $\mathbb{S}_y$ for class $y$ in a target task, we generate a set of feature vectors with label $y$ by sampling from the calibrated Gaussian distributions:

$$\mathbb{D}_y = \{(\boldsymbol{x}, y) | \boldsymbol{x} \sim \mathcal{N}(\boldsymbol{\mu}, \boldsymbol{\Sigma}), \forall (\boldsymbol{\mu}, \boldsymbol{\Sigma}) \in \mathbb{S}^y\}. \tag{7}$$

Here, the total number of generated features per class is set as a hyperparameter and they are equally distributed for every calibrated distribution in $\mathbb{S}_y$. The generated features along with the original support set features for a few-shot task is then served as the training data for a task-specific classifier. We train the classifier for a task by minimizing the cross-entropy loss over both the features of its support set $\mathcal{S}$ and the generated features $\mathbb{D}_y$:

$$\ell = \sum_{(\boldsymbol{x}, y) \sim \tilde{\mathcal{S}} \cup \mathbb{D}_y, y \in \mathcal{Y}^{\mathcal{T}}} -\log \Pr(y | \boldsymbol{x}; \theta), \tag{8}$$

where $\mathcal{Y}^{\mathcal{T}}$ is the set of classes for the task $\mathcal{T}$. $\tilde{\mathcal{S}}$ denotes the support set with features transformed by Turkey's Ladder of Powers transformation and the classifier model is parameterized by $\theta$.

## 4 EXPERIMENTS

In this section, we answer the following questions:

- How does our distribution calibration strategy perform compared to the state-of-the-art methods?

Table 3: 5way1shot and 5way5shot classification accuracy (%) on *tiered*ImageNet (Ren et al., 2018). The numbers in **bold** have intersecting confidence intervals with the most accurate method.

| Methods | *tiered*ImageNet | |
| --- | --- | --- |
| | 5way1shot | 5way5shot |
| Matching Net (Vinyals et al. (2016)) | $68.50 \pm 0.92$ | $80.60 \pm 0.71$ |
| Prototypical Net (Snell et al. (2017)) | $65.65 \pm 0.92$ | $83.40 \pm 0.65$ |
| LEO (Rusu et al. (2019)) | $66.33 \pm 0.05$ | $82.06 \pm 0.08$ |
| E3BM (Liu et al. (2020c)) | $71.20 \pm 0.40$ | $85.30 \pm 0.30$ |
| DeepEMD (Zhang et al., 2020) | $71.16 \pm 0.87$ | $86.03 \pm 0.58$ |
| Maximum Likelihood with DC (Ours) | $75.92 \pm 0.60$ | $87.84 \pm 0.65$ |
| SVM with DC (Ours) | $\mathbf{77.93 \pm 0.12}$ | $\mathbf{89.72 \pm 0.37}$ |
| Logistic Regression with DC (Ours) | $\mathbf{78.19 \pm 0.25}$ | $\mathbf{89.90 \pm 0.41}$ |

- What does calibrated distribution look like? Is it an accurate approximation for this class?
- How does Tukey's Ladder of Power transformation interact with the feature generations? How important is each in relation to performance?

## 4.1 EXPERIMENTAL SETUP

### 4.1.1 DATASETS

We evaluate our distribution calibration strategy on *mini*ImageNet (Ravi & Larochelle (2017)), *tiered*ImageNet (Ren et al. (2018)) and CUB (Welinder et al. (2010)). *mini*ImageNet and *tiered*ImageNet have a brand range of classes including various animals and objects while CUB is a more fine-grained dataset that includes various species of birds. Datasets with different levels of granularity may have different distributions for their feature space. We want to show the effectiveness and generality of our strategy on all three datasets.

***mini*ImageNet** is derived from ILSVRC-12 dataset (Russakovsky et al., 2014). It contains 100 diverse classes with 600 samples per class. The image size is $84 \times 84 \times 3$. We follow the splits used in previous works (Ravi & Larochelle, 2017), which split the dataset into 64 base classes, 16 validation classes, and 20 novel classes.

***tiered*ImageNet** is a larger subset of ILSVRC-12 dataset (Russakovsky et al., 2014), which contains 608 classes sampled from hierarchical category structure. Each class belongs to one of 34 higher-level categories sampled from the high-level nodes in the ImageNet. The average number of images in each class is 1281. We use 351, 97, and 160 classes for training, validation, and test, respectively.

**CUB** is a fine-grained few-shot classification benchmark. It contains 200 different classes of birds with a total of 11,788 images of size $84 \times 84 \times 3$. Following previous works (Chen et al., 2019a), we split the dataset into 100 base classes, 50 validation classes, and 50 novel classes.

### 4.1.2 EVALUATION METRIC

We use the top-1 accuracy as the evaluation metric to measure the performance of our method. We report the accuracy on 5way1shot and 5way5shot settings for *mini*ImageNet, *tiered*ImageNet and CUB. The reported results are the averaged classification accuracy over 10,000 tasks.

### 4.1.3 IMPLEMENTATION DETAILS

For feature extractor, we use the WideResNet (Zagoruyko & Komodakis, 2016) trained following previous work (Mangla et al. (2020)). For each dataset, we train the feature extractor with base classes and test the performance using novel classes. Note that the feature representation is extracted from the penultimate layer (with a ReLU activation function) from the feature extractor, thus the values are all non-negative so that the inputs to Tukey's Ladder of Powers transformation in Equation 3 are valid. At the distribution calibration stage, we compute the base class statistics and transfer them to calibrate novel class distribution for each dataset. We use the LR and SVM implementation of scikit-learn (Pedregosa et al. (2011)) with the default settings. We use the same hyperparameter value for all datasets except for $\alpha$. Specifically, the number of generated features

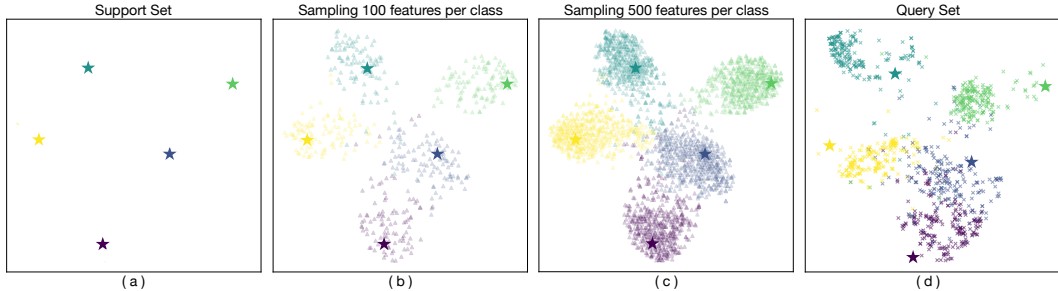

Figure 2: t-SNE visualization of our distribution estimation. Different colors represent different classes. '★' represents support set features, 'x' in figure (d) represents query set features, '▲' in figure (b)(c) represents generated features.

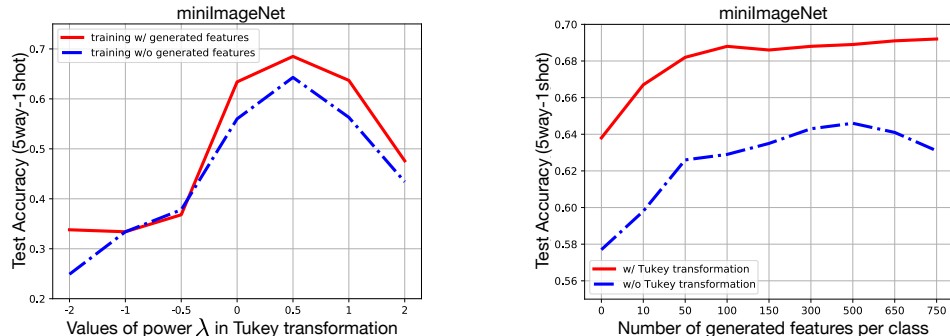

Figure 3: Left: Accuracy when increasing the power in Tukey's transformation when training with (red) or without (blue) the generated features. Right: Accuracy when increasing the number of generated features with the features are transformed by Tukey's transformation (red) and without Tukey's transformation (blue).

is 750; $k = 2$ and $\lambda = 0.5$. $\alpha$ is 0.21, 0.21 and 0.3 for miniImageNet, tieredImageNet and CUB, respectively.

## 4.2 COMPARISION TO STATE-OF-THE-ART

Table 2 and Table 3 presents the 5way1shot and 5way5shot classification results of our method on *mini*ImageNet, *tiered*ImageNet and CUB. We compare our method with the three groups of the few-shot learning method, *optimization-based*, *metric-based*, and *generation-based*. Our method can be built on top of any classifier, and we use two popular and simple classifiers, namely SVM and LR to prove the effectiveness of our method. Simple linear classifiers equipped with our method perform better than the state-of-the-art few-shot classification method and achieve the best performance on 1-shot and 5-shot settings of *mini*ImageNet, *tiered*ImageNet and CUB. The performance of our distribution calibration surpasses the state-of-the-art *generation-based* method by 10% for the 5way1shot setting, which proves that our method can handle extremely low-shot classification tasks better. Compared to other *generation-based* methods, which require the design of a generative model with extra training costs on the learnable parameters, simple machine learning classifier with DC is much more simple, effective and flexible and can be equipped with any feature extractors and classifier model structures. Specifically, we show three variants, i.e, Maximum likelihood with DC, SVM with DC, Logistic Regression with DC in Table 2 and Table 3. A simple maximum likelihood classifier based on the calibrated distribution can outperform previous baselines and training a SVM classifier or Logistic Regression classifier using the samples from the calibrated distribution can further improve the performance.

## 4.3 VISUALIZATION OF GENERATED SAMPLES

We show what the calibrated distribution looks like by visualizing the generated features sampled from the distribution. In Figure 2, we show the t-SNE representation (van der Maaten & Hinton

Table 4: Ablation study on *mini*ImageNet 5way1shot and 5way5shot showing accuracy (%) with 95% confidence intervals.

| Tukey transformation | Training with generated features | *mini*ImageNet | |
|:---:|:---:|:---:|:---:|
| | | 5way1shot | 5way5shot |
| ✗ | ✗ | $56.37 \pm 0.68$ | $79.03 \pm 0.51$ |
| ✓ | ✗ | $64.30 \pm 0.53$ | $81.33 \pm 0.35$ |
| ✗ | ✓ | $63.70 \pm 0.38$ | $82.26 \pm 0.73$ |
| ✓ | ✓ | $\mathbf{68.57 \pm 0.55}$ | $\mathbf{82.88 \pm 0.42}$ |

Table 5: 5way1shot classification accuracy (%) on *mini*ImageNet with different backbones.

| Backbones | without DC | with DC |
|:---|:---:|:---:|
| conv4 (Chen et al., 2019a) | $42.11 \pm 0.71$ | $\mathbf{54.62 \pm 0.64}$ (↑ **12.51**) |
| conv6 (Chen et al., 2019a) | $46.07 \pm 0.26$ | $\mathbf{57.14 \pm 0.45}$ (↑ **11.07**) |
| resnet18 (Chen et al., 2019a) | $52.32 \pm 0.82$ | $\mathbf{61.50 \pm 0.47}$ (↑ **9.180**) |
| WRN28 (Mangla et al., 2020) | $54.53 \pm 0.56$ | $\mathbf{64.38 \pm 0.63}$ (↑ **9.850**) |
| WRN28 + Rotation Loss (Mangla et al., 2020) | $56.37 \pm 0.68$ | $\mathbf{68.57 \pm 0.55}$ (↑ **12.20**) |

(2008)) of the original support set (a), the generated features (b,c) as well as the query set (d). Based on the calibrated distribution, the sampled features form a Gaussian distribution and more samples (c) can have a more comprehensive representation of the distribution. Due to the limited number of examples in the support set, only 1 in this case, the samples from the query set usually cover a greater area and are a mismatch with the support set. This mismatch can be fixed to some extent by the generated features, i.e., the generated features in (c) can overlap areas of the query set. Thus, training with these generated features can alleviate the mismatch between the distribution estimated only from the few-shot samples and the ground truth distribution.

## 4.4 APPLICABILITY OF DISTRIBUTION CALIBRATION

**Applying distribution calibration on different backbones**

Our distribution calibration strategy is agnostic to backbones / feature extractors. Table 5 shows the consistent performance boost when applying distribution calibration on different feature extractors, i.e, four convolutional layers (conv4), six convolutional layers (conv6), resnet18, WRN28 and WRN28 trained with rotation loss. Distribution calibration achieves around 10% accuracy improvement compared to the backbones trained with different baselines.

**Applying distribution calibration on other baselines**

A variety of works can benefit from training with the features generated by our distribution calibration strategy. We apply our distribution calibration strategy on two simple few-shot classification algorithms, Baseline (Chen et al., 2019a) and Baseline++ (Chen et al., 2019a). Table 6 shows that our distribution calibration brings over 10% of accuracy improvement on both.

## 4.5 EFFECTS OF FEATURE TRANSFORMATION AND TRAINING WITH GENERATED FEATURES

**Ablation Study**

Table 4 shows the performance when our model is trained without Tukey's Ladder of Powers transformation for the features as in Equation 3 and when it is trained without the generated features as in Equation 7. It is clear that there is a severe decline in performance of over 10% if both are not used in the 5way1shot setting. The ablation of either one results in a performance drop of around 5% in the 5way1shot setting.

**Choices of Power for Tukey's Ladder of Powers Transformation**

The left side of Figure 3 shows the 5way1shot accuracy when choosing different powers for the Tukey's transformation in Equation 3 when training the classifier with the generated features (red) and without (blue). Note that when the power $\lambda$ equals 1, the transformation keeps the original feature representations. There is a consistent general tendency for training with and without the

Table 6: 5way1shot classification accuracy (%) on *mini*ImageNet with different baselines using distribution calibration.

| Method | without DC | with DC |
|---|---|---|
| Baseline (Chen et al., 2019a) | $42.11 \pm 0.71$ | $\mathbf{54.62 \pm 0.64}$ (↑ **12.51**) |
| Baseline++ (Chen et al., 2019a) | $48.24 \pm 0.75$ | $\mathbf{61.24 \pm 0.37}$ (↑ **13.00**) |

generated features and in both cases, we found $\lambda = 0.5$ is the optimum choice. With the Tukey's transformation, the distribution of query set features in target tasks become more aligned to the calibrated Gaussian distribution, thus benefits the classifier which is trained on features sampled from the calibrated distribution.

**Number of generated features**

The right side of Figure 3 analyzes whether more generated features results in consistent improvement in both cases, namely when the features of support and query set are transformed by Tukey's transformation (red) and when they are not (blue). We found that when the number of generated features is below 500, both cases can benefit from more generated features. However, when more features are sampled, the performance of the classifier tested on untransformed features begins to decline. By training with the generated samples, the simple logistic regression classifier has a 12% relative performance improvement in a 1-shot classification setting.

### 4.6 OTHER HYPER-PARAMETERS

We select the hyperparameters based on the performance of the validation set. The $k$ base class statistics to calibrate the novel class distribution in Equation 5 is set to 2. Figure 4 shows the effect of different values of $k$. The $\alpha$ in Equation 6 is a constant added on each element of the estimated covariance matrix, which can determine the degree of dispersion of features sampled from the calibrated distributions. An appropriate value of $\alpha$ can ensure a good decision boundary for the classifier. Different datasets have different statistics and an appropriate value of $\alpha$ may vary for different datasets. Figure 5 explores the effect of $\alpha$ on all three datasets, i.e. *mini*ImageNet, *tiered*ImageNet and CUB. We observe that in each dataset, the performance of the validation set and the novel (testing) set generally has the same tendency, which indicates that the variance is dataset-dependent and is not overfitting to a specific set.

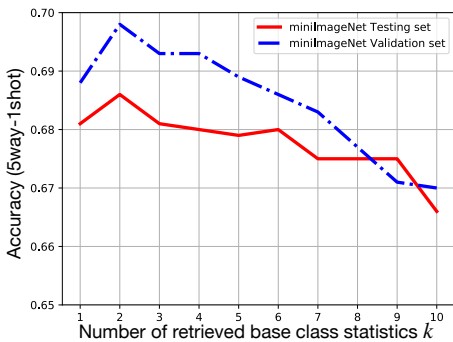

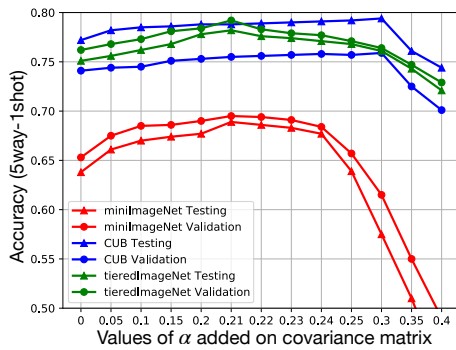

Figure 4: The effect of different values of $k$.   Figure 5: The effect of different values of $\alpha$.

## 5 CONCLUSION AND FUTURE WORKS

We propose a simple but effective distribution calibration strategy for few-shot classification. Without complex generative models, training loss and extra parameters to learn, a simple logistic regression trained with features generated by our strategy outperforms the current state-of-the-art methods by $\sim 5\%$ on *mini*ImageNet. The calibrated distribution is visualized and demonstrates an accurate estimation of the feature distribution. Future works will explore the applicability of distribution calibration on more problem settings, such as multi-domain few-shot classification, and more methods, such as metric-based meta-learning algorithms.

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

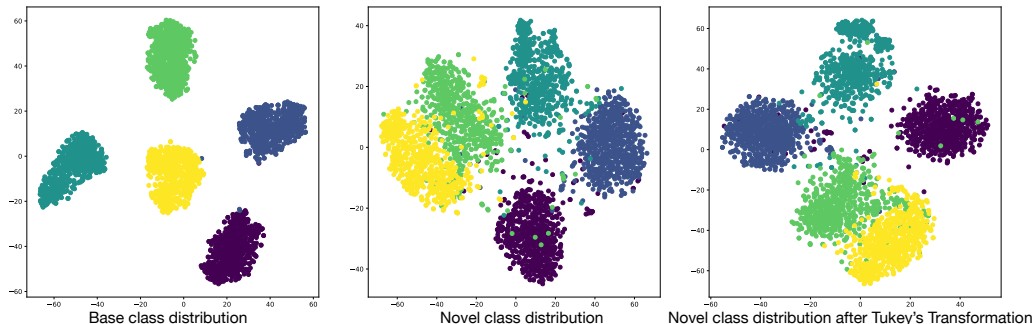

Figure 6: We show the feature distribution of 5 base classes, and the feature distribution of 5 novel classes before/after Tukey's Transformation.

| Training data | miniImageNet 5way1shot |
|---|---|
| Support set only | $56.37 \pm 0.68$ |
| Support set + 1 feature from the nearest class | $62.39 \pm 0.49$ |
| Support set + 5 features from the nearest class | $59.73 \pm 0.42$ |
| Support set + 10 features from the nearest class | $58.93 \pm 0.49$ |
| Support set + 100 features from the nearest class | $57.33 \pm 0.48$ |
| Support set + 100 features sampled from calibrated distribution | $\mathbf{68.53 \pm 0.32}$ |

Table 7: The comparison with nearest class feature augmentation.

## A  AUGMENTATION WITH NEAREST CLASS FEATURES

Instead of sampling from the calibrated distribution, we can simply retrieve examples from the nearest class to augment the support set. Table 7 shows the comparison of training using samples from the calibrated distribution, the different number of retrieved features from the nearest class, and only using the support set. We found the retrieved features can improve the performance compared to only using the support set but can damage the performance when increasing the number of retrieved features, where the retrieved samples probably serve as noisy data for tasks targeting different classes.

## B  DISTRIBUTION CALIBRATION WITHOUT NOVEL FEATURE

We calibrate the novel class mean by averaging the novel class mean and the retrieved base class means in Equation 6. Table 8 shows the distribution calibration without averaging novel feature, in which the calibrated mean is calculated as $\boldsymbol{\mu}' = \frac{\Sigma_{i \in \mathbb{S}_N} \boldsymbol{\mu}_i}{k}$.

## C  THE EFFECTS OF TUKEY'S TRANSFORMATION

Figure 6 shows the distribution of 5 base classes and 5 novel classes before/after Tukey's transformation. It is observed that the base class distribution satisfies Gaussian assumption well (left) while the novel class distribution is more skew (middle). The novel class distribution after Tukey's transformation (right) is more aligned with the Gaussian-like base class distribution.

| | miniImageNet 5way1shot |
|---|---|
| Distribution Calibration w/o novel feature $\tilde{\boldsymbol{x}}$ | $59.38 \pm 0.73$ |
| Distribution Calibration w/ novel feature $\tilde{\boldsymbol{x}}$ | $\mathbf{68.57 \pm 0.55}$ |

Table 8: The comparison between distribution calibration with and without novel feature $\tilde{\boldsymbol{x}}$.

| Novel class | Top-1 base class similarity | Top-2 base class similarity | DC improvement |
|---|---|---|---|
| malamute | 93% | 85% | ↑ 21.30% |
| golden retriever | 85% | 74% | ↑ 18.37% |
| ant | 71% | 67% | ↑ 9.77% |

Table 9: Performance improvement with respect to the similarity level between a query novel class and the most similar base classes.

## D  THE SIMILARITY LEVEL ANALYSIS

We found that the higher similarities between the retrieved base class distribution and the novel class ground-truth distribution, the higher the performance improvement our method will bring as shown in Table 9. The results in the table are under 5-way-1-shot setting.

