# OpenReview forum: "Free Lunch for Few-shot Learning:  Distribution Calibration"
_ICLR.cc/2021/Conference — ICLR 2021 Oral_

### Official Review · AnonReviewer1 · 2020-10-28
**A simple data augmentation for few-shot image classification**

**Rating:** 7
**Confidence:** 4

**Review:**

This paper presents a simple and intuitive data augmentation method for few shot image classification. The proposed method assumes the feature distribution of a single class to be gaussian. Based on this assumption, the proposed method samples features from a gaussian distribution that is constructed using the statistics of the similar base classes. Combined with logistic regression, the proposed method achieves strong performance on two standard few-shot image classification benchmarks. This submission also carries out an ablation study on several design choices, which I really appreciate.

The submission is well-written and clear. The proposed method is novel and can inspire future augmentation-based methods in few-shot image classification.

I have a few more requests/ablations that I am curious to see.

- Looking at Figure 4, the 1-shot accuracy with only 1 retrieved class is already very strong. Instead of sampling from the "calibrated" distribution, can we simply retrieve examples from the nearest class? Namely, find the nearest class, randomly sample some examples, and use their features to augment the novel classes. This ablation should make clear what additional value the sampling procedure adds.
- In equation (6), how important is it to include the novel class feature into the mean? Can we simply do \mu_prime = \sum_{i \in S_N} \mu_i /  k?
- This method makes an important assumption that the feature distribution is gaussian. How well does this assumption hold in practice? Can the authors provide some analysis of the feature distribution? Ideally both before and after Tukey’s Ladder of Powers transformation.

---

> ### Author Response · Authors · 2020-11-17
> **Response to AnonReviewer1**
>
>
>
>
> Thanks for taking the time to review our paper! We are happy to respond to your comments and questions as below:
>
> **Q1:Looking at Figure 4, the 1-shot accuracy with only 1 retrieved class is already very strong. Instead of sampling from the "calibrated" distribution, can we simply retrieve examples from the nearest class? Namely, find the nearest class, randomly sample some examples, and use their features to augment the novel classes. This ablation should make clear what additional value the sampling procedure adds.**
>
> A1:
> Thanks for this comment! We have included the comparison of training using samples from the calibrated distribution, different number of retrieved features from nearest class, and only using support set in Table 7 (in Appendix) as shown below:
>
> | Training data &nbsp;|&nbsp; accuracy on miniImageNet 5way1shot |
> |:--|:--:|
> | Support set only &nbsp;|&nbsp; 56.37 $\pm$ 0.68 |
> | Support set + 1 feature from the nearest class&nbsp;|&nbsp;62.39 $\pm$ 0.49|
> | Support set + 5 features from the nearest class&nbsp;|&nbsp;59.73 $\pm$ 0.42|
> | Support set + 10 features from the nearest class&nbsp;|&nbsp;58.93 $\pm$ 0.49|
> | Support set + 100 features from the nearest class&nbsp;|&nbsp;57.33 $\pm$ 0.48|
> | Suppport set + 100 features sampled from calibrated distribution (ours)&nbsp;|&nbsp;**68.53** $\pm$ **0.32**|
>
>
> We empirically found the retrieved features can improve the performance compared to only using the support set but can damage the performance when increasing the number of retrieved features, where the retrieved samples probably serve as noisy data for tasks targeting different classes.
>
>
> **Q2:In equation (6), how important is it to include the novel class feature into the mean? Can we simply do $\mu_{prime} = \sum_{i \in S_N} \mu_i / k$?**
>
> A2:
> Thanks for this comment! We have added Table 8 (in Appendix) as below:
>
> |  &nbsp;|&nbsp; miniImageNet 5way1shot |
> |:----|:----:|
> |Distribution calibration w/o novel feature &nbsp;|&nbsp; 59.38 $\pm$ 0.73|
> | Distribution calibration w/ novel feature &nbsp;|&nbsp; **68.57** $\pm$ **0.55** |
>
> We found the support set is essential for performance when calibrating the distribution.
>
> **Q3:This method makes an important assumption that the feature distribution is gaussian. How well does this assumption hold in practice? Can the authors provide some analysis of the feature distribution? Ideally both before and after Tukey's Ladder of Powers transformation.**
>
> A3:
> Thanks for this suggestion! We have included the feature distribution of 5 randomly sampled base classes and the feature distribution of 5 randomly sampled novel classes before/after Tukey's transformation in Figure 6 (in Appendix).
> The base classes' feature distributions satisfy the Gaussian assumption well (left) while the feature distributions of novel classes are more skewed (middle). After transformation (right), the calibrated novel class distributions are more aligned with the Gaussian-like base feature distribution.
>
> We hope our responses address your concerns and happy to continue the discussion if there are any other questions!

---

### Official Review · AnonReviewer3 · 2020-10-28
**Simple and effective method for calibrating the few-shot class distribution**

**Rating:** 7
**Confidence:** 4

**Review:**

Summary:

This paper identifies the problem of biased distributions in few-shot learning and proposes to fix it. In few-shot learning, only a few samples per class are available; this makes estimating the class distribution difficult. The paper proposes a distribution calibration algorithm that makes use of the meta-train class distributions to calibrate the few-shot class distributions. Once calibrated, more samples are drawn from this distribution to learn a classifier that generalizes better. This approach does not require additional learnable parameters and can be (potentially) built on-top of any pre-trained feature extractor. Empirical results show that this approach achieves state-of-the-art results on Mini-ImageNet and CUB.

Pros:
1. This paper identifies and tries to tackle an important problem in few-shot learning - estimation of the class distribution. Due to the limited number of samples, this problem is difficult, but important for few-shot learning. The proposed algorithm is simple and effective in tacking this problem.
2. As opposed to other related works, the proposed algorithm does not have any learnable parameters. It makes use of the features obtained for the meta-train and few-shot samples.

Cons:
1. One of the claims of the paper is that the proposed algorithm is pre-trained feature extractor agnostic. However, there are no experiments to validate this claim. Consider adding feature extractors trained in different ways.
2. Some of the popular few-shot datasets were not included in the experimental section, namely Tiered-ImageNet [1] and Meta-Dataset [2]. The former is a larger dataset than the ones used, and the latter provides cross-domain results for the approach. The proposed algorithm assumes that statistics from the meta-train classes would transfer to the few-shot classes, which would be tested more thoroughly in the cross-domain setting.

Clarifications:
1. In the feature space, it is intuitive that the means of similar classes would be correlated. Is there a justification for why this is true for the variances?
2. Tukey's Ladder of Powers transformation is used only on the few-shot samples and not the meta-train samples. Is there a reason for that? If the pre-trained feature extractor is trained using a certain metric (cosine distances for example), I would imagine transforming all the features would be beneficial.
3. The backbone used in the experiments is trained using a supervised and self-supervised loss. What are the results without the self-supervised loss?
4. How many samples are drawn from the calibrated distribution for the numbers in Table 2?

Notes:
1. The performance of few-shot learning algorithms has been traditionally evaluated using the averaged accuracy over multiple tasks, but that is not the only way to do it. Look at [3] for details.

[1] Mengye Ren et al. Meta-Learning for Semi-Supervised Few-Shot Classification.
[2] Eleni Triantafillou et al. Meta-Dataset: A Dataset of Datasets for Learning to Learn from Few Examples.
[3] Guneet S. Dhillon et al. A Baseline for Few-Shot Image Classification.

---

> ### Author Response · Authors · 2020-11-17
> **Response to AnonReviewer3**
>
>
> Thanks for taking the time to review our paper! We are happy to respond to your comments and questions as below:
>
> **Q1: One of the claims of the paper is that the proposed algorithm is pre-trained feature extractor agnostic. However, there are no experiments to validate this claim. Consider adding feature extractors trained in different ways.**
>
> A1:
> Thanks for your comment! We have updated the paper and add the results on various backbones as Table 5. We show significant and consistent improvement on various backbones structures. A summary of the results are as below:
>
> | Backbones  &nbsp;|&nbsp; without DC &nbsp;|&nbsp; with DC |
> |:--|:--:|:--:|
> | conv4 &nbsp;|&nbsp; 42.11 $\pm$ 0.71 &nbsp;|&nbsp; **54.62** $\pm$ **0.64** (↑ **12.51**)|
> | conv6 &nbsp;|&nbsp; 46.07 $\pm$ 0.26 &nbsp;|&nbsp; **57.14** $\pm$ **0.45**(↑ **11.07**)|
> | resnet18 &nbsp;|&nbsp; 52.32 $\pm$ 0.82 &nbsp;|&nbsp;**61.50** $\pm$ **0.47** (↑ **9.18**)|
> | WRN28 &nbsp;|&nbsp; 54.53 $\pm$ 0.56 &nbsp;|&nbsp;**64.38** $\pm$ **0.63** (↑ **9.85**)|
> | WRN28+Rotation Loss &nbsp;|&nbsp; 56.37 $\pm$ 0.68 &nbsp;|&nbsp;**68.57** $\pm$ **0.55** (↑ **12.20**)|
>
> **Q2: Some of the popular few-shot datasets were not included in the experimental section, namely Tiered-ImageNet [1] and Meta-Dataset [2]. The former is a larger dataset than the ones used, and the latter provides cross-domain results for the approach. The proposed algorithm assumes that statistics from the meta-train classes would transfer to the few-shot classes, which would be tested more thoroughly in the cross-domain setting.**
>
> A2:
> Thanks for this suggestion! We have included the result on tieredImageNet as Table 3 . We are happy to see our method can also set new state-of-the-art results on tieredImageNet! Here is a summary of the comparison:
>
> | Methods &nbsp;|&nbsp; 5way1shot &nbsp;|&nbsp; 5way5shot|
> |:--|:--:|:--:|
> | Matching Net[A] &nbsp;|&nbsp; 68.50 $\pm$ 0.92 &nbsp;|&nbsp; 80.60 $\pm$ 0.71|
> |Prototypical Net[B]&nbsp;|&nbsp;65.65 $\pm$ 0.92 &nbsp;|&nbsp;83.40 $\pm$ 0.65|
> |LEO[C] &nbsp;|&nbsp;66.33 $\pm$ 0.05&nbsp;|&nbsp; 82.06 $\pm$ 0.08|
> |E3BM[D]&nbsp;|&nbsp; 71.20 $\pm$ 0.40&nbsp;|&nbsp; 85.30 $\pm$ 0.30|
> |Distribition Calibration (Ours)&nbsp;|&nbsp; **78.19** $\pm$ **0.25** &nbsp;|&nbsp;**89.90** $\pm$ **0.41**|
>
>
>
> **Q3: In the feature space, it is intuitive that the means of similar classes would be correlated. Is there a justification for why this is true for the variances?**
>
> A3:
> We empirically found this is also true for the variance. This is also found in [E], in which a hierarchical Bayesian model is learned to build a prior over category means and variance. Please refer to Figure 3 in their paper for a display of mean and variance.
>
> **Q4: Tukey's Ladder of Powers transformation is used only on the few-shot samples and not the meta-train samples. Is there a reason for that? If the pre-trained feature extractor is trained using a certain metric (cosine distances for example), I would imagine transforming all the features would be beneficial.**
>
> A4:
> Thanks for pointing it out! We have tried applying the transformation on both meta-trained samples and samples from test tasks. We found the performance is slightly lower as below:
>
> |  &nbsp;|&nbsp; miniImageNet 5way1shot |
> |--|--|
> | Transform meta-train and novel features &nbsp;|&nbsp;  67.34 $\pm$ 0.73  |
> | Transform novel features&nbsp;|&nbsp; **68.57** $\pm$ **0.55** |
>
> A possible reason is that the meta-trained samples have already been in the shape of Gaussian due to sufficient training samples as shown in the left of Figure 6  (in Appendix) in our updated paper. Thus, transformation on the meta-trained samples is not necessary.
>
> **Q5: The backbone used in the experiments is trained using a supervised and self-supervised loss. What are the results without the self-supervised loss?**
>
> A5:
> We show the performance improvement in both cases below:
>
> | Backbones  &nbsp;|&nbsp; without DC &nbsp;|&nbsp; with DC |
> |:--|:--:|:--:|
> | without self-supervised loss &nbsp;|&nbsp; 54.53 $\pm$ 0.56 &nbsp;|&nbsp;**64.38** $\pm$ **0.63** |
> | with self-supervised loss &nbsp;|&nbsp; 56.37 $\pm$ 0.68 &nbsp;|&nbsp;**68.57** $\pm$ **0.55**|
>
> **Q6: How many samples are drawn from the calibrated distribution for the numbers in Table 2?**
>
> A6:
>  750 samples per class. Thanks for this comment, and we have included it in Section 4.1.3.
>
>  We really hope our response addresses your concern. If you have any other questions, we are very happy to continue discussions!
>
>
> [A] Vinyals et al., Matching Networks for One Shot Learning, NeurlPS 2016
> [B] Snell et al., Prototypical Networks for Few-shot Learning, NeurlPS 2017
> [C] Rusu et al., Meta-Learning with Latent Embedding Optimization, ICLR 2019
> [D] Liu et al., An Ensemble of Epoch-wise Empirical Bayes for Few-shot Learning, ECCV 2020
> [E] Salakhutdinov et al., One-Shot Learning with a Hierarchical Nonparametric Bayesian Model, ICML Workshop 2012

---

### Official Review · AnonReviewer4 · 2020-10-31
**Marginally below threshold.**

**Rating:** 7
**Confidence:** 5

**Review:**

Summary

The paper proposes a method to calibrate the underlying distribution of a few samples in the few-shot classification scenario. The idea is to estimate a feature distribution of a few samples of a novel class from base class distributions. The authors assume that every dimension in the feature vector follows a Gaussian distribution. Based on the observation that the mean and variance of the distribution with respect to each class are correlated to the semantic similarity of each class, base class distribution can be transferred to the novel class distribution. After distribution calibration, features can be generated from the calibrated distribution and the generated features are used to train classifiers. SVM and logistic regression classifier are used to verify the approach on the mini-imagenet and CUB datasets.

Pros
-   The idea can be applied to any types of feature extractors or classifiers when the Gaussian distribution assumption holds.
-   The proposed approach shows competitive performance on two benchmarks, mini-imagenet and CUB.
-   Extensive ablation studies verify hyper-parameter settings and their sensitivities.

Cons
- Many hyper-parameters are involved in the approach. ( Turkey’s transformation parameter lambda, top-k base distributions, dispersion parameter alpha, number of generated features)
-   Some hyper-parameter setting should be different depending on the dataset. (dispersion parameter alpha)
-   The paper uses the previous work (Mangla et al. 2020) as a baseline and applies their approach on top of it. While theoretically the approach can be applied to any types of feature extractors, only one baseline improvement is shown in the paper.


Rating

I give marginally below the threshold. The paper shows good performance and also claims the approach can be applied on top of any classifiers or feature extractor. The general applicability and effectiveness are the strength of the paper. However, the proposed approach is only verified on one baseline approach (Mangla et al. 2020). More empirical data would strengthen the claim of the paper.

Questions

The approach outputs calibrated distribution, i.e., a mixture of Gaussian distributions. These distributions can be directly used to classify query samples by calculating the likelihood of the sample on each distribution. How does likelihood classification perform compared to the retrained classifiers (SVM or RL)?
The optimal values for top-k parameter and alpha parameter are different depending on the dataset. Are optimal values for lambda and the number of generated features same or different depending on the dataset? Did the authors use the same lambda and number of generated features for both mini-imagenet and CUB experiments?
The authors only applied the proposed method on one baseline approach. Is the approach effective for more variety of backbone networks and losses?

Feedback

- Table1 shows that there is a correlation between feature distributions and semantic similarities. It would be interesting to see how the distribution calibration performs on each class depending on the similarity level of top-k base distributions.
- The empirical evidence is not sufficient to claim the approach applies to general network architectures and few-shot learning approaches. Experiments with more baseline approach to show the effectiveness of the idea is recommended.

---

> ### Author Response · Authors · 2020-11-17
> **Response to AnonReviewer4 (1/2)**
>
> We appreciate the thorough review of R4. We found your main concerns are the applicability of our model and clarifications on hyper-parameters. Please find our responses below:
>
> **Q1: The paper uses the previous work (Mangla et al. 2020) as a baseline and applies their approach on top of it. While theoretically the approach can be applied to any types of feature extractors, only one baseline improvement is shown in the paper.**
>
> A1:
> Thanks for this suggestion! We have included more experiment results to show our model's applicability and robustness from three aspects: Distribution Calibration (DC) on different backbones, baselines, and datasets.
> 1. DC on different backbones.
> We show DC can significantly and consistently improve the performance when applying different backbone structures, as shown in Table 5:
>
> | Backbones  &nbsp;|&nbsp; without DC &nbsp;|&nbsp; with DC |
> |:--|:--:|:--:|
> | conv4 &nbsp;|&nbsp; 42.11 $\pm$ 0.71 &nbsp;|&nbsp; **54.62** $\pm$ **0.64** (↑ **12.51**)|
> | conv6 &nbsp;|&nbsp; 46.07 $\pm$ 0.26 &nbsp;|&nbsp; **57.14** $\pm$ **0.45**(↑ **11.07**)|
> | resnet18 &nbsp;|&nbsp; 52.32 $\pm$ 0.82 &nbsp;|&nbsp;**61.50** $\pm$ **0.47** (↑ **9.18**)|
> | WRN28 &nbsp;|&nbsp; 54.53 $\pm$ 0.56 &nbsp;|&nbsp;**64.38** $\pm$ **0.63** (↑ **9.85**)|
> | WRN28+Rotation Loss &nbsp;|&nbsp; 56.37 $\pm$ 0.68 &nbsp;|&nbsp;**68.57** $\pm$ **0.55** (↑ **12.20**)|
>
> 2. DC on different baselines.
> We show DC can also improve over previous few-shot learning baselines when training using the sampled features from DC as in Table 6:
>
> | Methods &nbsp;|&nbsp; without DC  &nbsp;|&nbsp; with DC |
> |:--|:--:|:--:|
> | Baseline[E] &nbsp;|&nbsp; 42.11 $\pm$ 0.71 &nbsp;|&nbsp; **54.62** $\pm$ **0.64** (↑ **12.51**)|
> | Baseline++[E]&nbsp;|&nbsp;48.24 $\pm$ 0.75 &nbsp;|&nbsp; **61.24** $\pm$ **0.37** (↑ **13.00**)|
>
> Also, DC is applicable to different traditional machine learning classifiers, i.e., maximum likelihood classifier, SVM and Logistic Regression as shown in Table 2 and Table 3:
>
> | Methods  &nbsp;|&nbsp; miniImageNet 1-shot&nbsp;|&nbsp; tieredImageNet 1-shot|
> |:--|:--:|:--:|
> | E3BM[D] \(previous sota\) &nbsp;|&nbsp; 63.80 $\pm$ 0.40 &nbsp;|&nbsp; 71.2 $\pm$ 0.40 |
> |Maximum Likelihood with DC&nbsp;|&nbsp; 66.91 $\pm$ 0.17&nbsp;|&nbsp; 75.92 $\pm$ 0.60|
> |SVM with DC&nbsp;|&nbsp; 67.31 $\pm$ 0.83&nbsp;|&nbsp; 77.93 $\pm$ 0.12|
> |Logistic Regression with DC&nbsp;|&nbsp; **68.57** $\pm$ **0.55**&nbsp;|&nbsp; **78.19** $\pm$ **0.25**|
>
> 3. DC on a more large-scaled dataset.
>
> Besides miniImageNet and CUB, we also benchmark on tieredImageNet, as in Table 3. We also set the new state-of-the-art on tieredImageNet.
>
> | Methods &nbsp;|&nbsp; 5way1shot &nbsp;|&nbsp; 5way5shot|
> |:--|:--:|:--:|
> | Matching Net[A] &nbsp;|&nbsp; 68.50 $\pm$ 0.92 &nbsp;|&nbsp; 80.60 $\pm$ 0.71|
> |Prototypical Net[B]&nbsp;|&nbsp;65.65 $\pm$ 0.92 &nbsp;|&nbsp;83.40 $\pm$ 0.65|
> |LEO[C] &nbsp;|&nbsp;66.33 $\pm$ 0.05&nbsp;|&nbsp; 82.06 $\pm$ 0.08|
> |E3BM[D]&nbsp;|&nbsp; 71.20 $\pm$ 0.40&nbsp;|&nbsp; 85.30 $\pm$ 0.30|
> |Distribition Calibration (Ours)&nbsp;|&nbsp; **78.19** $\pm$ **0.25** &nbsp;|&nbsp;**89.90** $\pm$ **0.41**|
>
> **Q2: Many hyper-parameters are involved in the approach. (Turkey's transformation parameter lambda, top-k base distributions, dispersion parameter alpha, number of generated features). Some hyper-parameter settings should be different depending on the dataset. (dispersion parameter alpha)**
>
> A2:
> Thanks for this comment! As shown in Figure 3, Figure 4 and Figure 5, as well as Section 4.5 and 4.6, we have thoroughly analyzed how these four hyperparameters will affect the performance. In sum, we found the selection of these hyperparameters are not dataset-dependent (we use the same value for three datasets), except for the dispersion parameter alpha. This is because different datasets have different Gaussian distribution of their features and setting different dispersions can better approximate the ground truth distribution.
> We have added related clarifications in Section 4.1.3 in our updated version of the submission! The values of all hyper-parameters on three datasets are as below:
>
> | Hyper-parameter &nbsp;|&nbsp; miniImageNet &nbsp;|&nbsp; tieredImageNet &nbsp;|&nbsp; CUB|
> |:--|:--:|:--:|:--:|
> | $k$ &nbsp;|&nbsp; 2 &nbsp;|&nbsp; 2&nbsp;|&nbsp; 2|
> | $\lambda$&nbsp;|&nbsp;0.5&nbsp;|&nbsp;0.5&nbsp;|&nbsp;0.5|
> |number of generated features&nbsp;|&nbsp;750&nbsp;|&nbsp;750&nbsp;|&nbsp;750|
> |$\alpha$&nbsp;|&nbsp;0.21&nbsp;|&nbsp;0.21&nbsp;|&nbsp;0.3|

---

> > ### Author Response · Authors · 2020-11-17
> > **Response to AnonReviewer4 (2/2)**
> >
> > **Q3: How does likelihood classification perform compared to the retrained classifiers (SVM or LR)?**
> >
> > A3:
> > Thanks for this comment! We have included the result using maximum likelihood in Table2 and Table 3 as below:
> >
> > | Methods  &nbsp;|&nbsp; miniImageNet 1-shot&nbsp;|&nbsp; tieredImageNet 1-shot|
> > |:--|:--:|:--:|
> > |Maximum Likelihood with DC&nbsp;|&nbsp; 66.91 $\pm$ 0.17&nbsp;|&nbsp; 75.92 $\pm$ 0.60|
> > |SVM with DC&nbsp;|&nbsp; 67.31 $\pm$ 0.83&nbsp;|&nbsp; 77.93 $\pm$ 0.12|
> > |Logistic Regression with DC&nbsp;|&nbsp; **68.57** $\pm$ **0.55**&nbsp;|&nbsp; **78.19** $\pm$ **0.25**|
> >
> > We found it can achieve competitive performance while training a SVM / LR classifier using the samples from the calibrated distribution can further improve the performance.
> >
> > **Q4: Table 1 shows that there is a correlation between feature distributions and semantic similarities. It would be interesting to see how the distribution calibration performs on each class depending on the similarity level of top-k base distributions.**
> >
> > A4:
> > Thanks for this comment!
> > We found that the higher similarities between the retrieved base class distribution and the novel class ground-truth distribution, the higher the performance improvement our method will bring as shown in Table 9:
> >
> > |  Novel class         &nbsp;|&nbsp;   Top-1 base class similarity &nbsp;|&nbsp;Top-2 base class similarity   &nbsp;|&nbsp; DC improvement |
> > |:----|:----:|:----:|:----:|
> > | malamute &nbsp;|&nbsp; 93\% &nbsp;|&nbsp; 85\% &nbsp;|&nbsp; ↑21.30\% acc|
> > | golden retriever    &nbsp;|&nbsp;   85\% &nbsp;|&nbsp; 74\% &nbsp;|&nbsp; ↑18.37\% acc  |
> > | ant| 71\% &nbsp;|&nbsp; 67\% &nbsp;|&nbsp;↑9.77\% acc|
> >
> > We really hope our response addresses your concern. If you have any other questions, we are very happy to continue discussions!
> >
> > [A] Vinyals et al., Matching Networks for One Shot Learning, NeurlPS 2016
> > [B] Snell et al., Prototypical Networks for Few-shot Learning, NeurlPS 2017
> > [C] Rusu et al., Meta-Learning with Latent Embedding Optimization, ICLR 2019
> > [D] Liu et al., An Ensemble of Epoch-wise Empirical Bayes for Few-shot Learning, ECCV 2020
> > [E] Chen et al., A Closer Look at Few-shot Classification, ICLR 2019

---

### Author Response · Authors · 2020-11-19
**Summary of Revisions**

We really appreciate all three reviewers for their valuable comments.
We’ve uploaded a revised draft incorporating reviewer feedback. Below is a summary of the main changes:

- The experiments on various backbones are included in Table 5. (R3 & R4)
- The experiments on various baseline models are included in Table 6. (R4)
- We also set new state-of-the-art performance on a larger dataset, tieredImageNet, in Table 3. (R3)
- We clarify the values of hyper-parameters on all three datasets in Section 4.1.3, and update the Figure 5 to provide more analysis. (R4)
- We visualize the feature distribution of randomly selected base classes and novel classes, as well as the distribution of novel classes after Tukey’s transformation in Figure 6 (in Appendix). (R1)

We really hope our responses and revisions address all reviewers’ concerns!

---

### Comment · ~Yulin_Wang1 · 2021-01-16
**A related work**

Hi authors,

Thanks for this interesting work for facilitating Few-shot Learning with data augmentation techniques in deep feature space. I'd like to point a (possibly) missing related work, which may be worthwhile to be discussed in the paper.

The implicit semantic data augmentation technique (ISDA, published in NeurIPS 2019, https://arxiv.org/abs/1909.12220) also performs data augmentation using deep features. They establish a zero-mean Gaussian distribution for each class and sample infinite semantic directions from it to augment training samples. The variance of the distribution is also estimated using intra-class Statistics. However, ISDA does not focus on Few-shot Learning and does not introduce techniques like Distribution Calibration like this paper.

I believe that citing ISDA won't devalue yours at all. If you want,  I will be happy to discuss further.

---

> ### Comment · ~Shuo_Yang5 · 2021-01-16
> **Thanks for pointing out the related work!**
>
> Hi Yulin,
>
> Thanks for pointing out your ISDA paper. It is an interesting data augmentation work and is very worthwhile to be discussed in our paper. We will add your work to our camera-ready version paper.
>
> Thank you!

---

> > ### Comment · ~Zhipeng_Zhou2 · 2021-04-22
> > **Maybe a related work is missing**
> >
> > Hi authors,
> >
> > Thanks for your great work. In fact, I have read one paper "Leveraging the Feature Distribution in Transfer-based Few-Shot Learning" which is posted on the arxiv half years ago, and it has almost the same idea as your paper. I wonder if this paper is the conference version of that preprint paper.  Or can you tell me the relationship or difference between these two versions?
> >
> > Thank you!

---

> > ### Comment · ~Jaekyeom_Kim1 · 2021-04-22
> > **Related work suggestion**
> >
> > Hi Shuo,
> >
> > Thanks for the great work and congratulations on getting an oral presentation session for your paper.
> > I'd suggest a potential related work, Model-Agnostic Boundary-Adversarial Sampling for Test-Time Generalization in Few-Shot Learning (2020, https://www.ecva.net/papers/eccv_2020/papers_ECCV/html/2307_ECCV_2020_paper.php).
> > It fine-tunes the feature extractor and thus the feature space solely in the test time by generating samples adversarial to classification boundaries, and it operates independently of training and without learning to create samples.
> > If you're willing to update your paper, it would be grateful to see our work discussed in it.
> >
> > Thank you!

---

### Decision · Program_Chairs · 2021-01-07
**Final Decision**

**Decision:**

Accept (Oral)

**Comment:**

This paper proposes a novel and powerful data augmentation strategy for few-shot learning, producing convincing improvements over current approaches. The request by the reviewers to include additional ablations, more backbones, and an additional dataset have been satisfactorily  resolved, with the results remaining strong. The reviewers are all unanimous in their recommendation that the paper be accepted for publication.